# Extreme potential photocatalysis enabled by spin-exchange Auger processes in magnetic-doped quantum dots

Qinxuan Cao[1], Jianning Feng [●][1], Kezhou Fan[2], Shuting Zhang[1], Jinzhong Zhang[3], Baixu Ma[4], Jie Xue[1], Xin Li [●][1], Kang Wang[3], Lizhi Tao [●][4], Aleksandr Sergeev[2], Ye Yang [●][3,5], Kam Sing Wong [●][2], Yong Huang [●][1] & Haipeng Lu [●][1,6,7] ✉

Visible-light-absorbing semiconductor nanocrystals have shown great promise as photocatalysts for promoting photoredox chemistry. However, their utilization in organic synthesis remains considerably limited compared to small molecule photosensitizers. Recently, the generation of hot electrons from quantum-confined systems has emerged as a powerful means of photoreduction, yet the efficiencies remain limited under mild conditions. In this study, we present an efficient hot-electron generation system facilitated by the spin-exchange Auger process in $Mn^{2+}$-doped CdS/ZnS quantum dots. These hot electrons can be effectively utilized in a wide range of organic reactions, such as the Birch reduction and reductive cleavage of C-Cl, C-Br, C-I, C-O, C-C, and N-S bonds. Notably, these reactions accommodate substrate reduction potentials as low as −3.4 V versus the saturated calomel electrode. Through two-photon excitation, we achieve the generation of a "super" photoreductant using visible-light irradiation power that is only 1% of that previously reported for molecular and quantum dot systems. By modulating the intensity of light output, the spin-exchange Auger process enables the on/off generation of hot electrons, allowing for programmable assembly-point cross-coupling cascades. Our findings demonstrate the potential of quantum-confined semiconductors in facilitating challenging organic transformations that were unattainable with molecular photocatalysts.

In recent years, colloidal semiconductor nanocrystals, commonly known as quantum dots (QDs), have gained significant attention as versatile photocatalysts for organic synthesis[1,2]. Unlike their traditional small-molecule counterparts, QDs exhibit superior photostability, higher absorption coefficients with broad and tunable absorption spectra, and extended carrier lifetimes. The charge transfer or energy transfer (ET) between QDs and organic substrates (Fig. 1a) has been reported to be remarkably rapid (~ps-ns) and

[1]Department of Chemistry, The Hong Kong University of Science and Technology, Clear Water Bay, Kowloon, 999077 Hong Kong, (SAR), China. [2]Department of Physics, The Hong Kong University of Science and Technology, Clear Water Bay, Kowloon, 999077 Hong Kong, (SAR), China. [3]State Key Laboratory of Physical Chemistry of Solid Surfaces, College of Chemistry and Chemical Engineering, Xiamen University, Xiamen 361005, China. [4]Department of Chemistry, Department of Chemistry, Southern University of Science and Technology, Shenzhen, Guangdong 518055, China. [5]Innovation Laboratory for Sciences and Technologies of Energy Materials of Fujian Province (IKKEM), Xiamen 361005, China. [6]Energy Institute, The Hong Kong University of Science and Technology, Clear Water Bay, Kowloon, 999077 Hong Kong, (SAR), China. [7]Hong Kong Branch of Chinese National Engineering Research Center for Tissue Restoration and Reconstruction, The Hong Kong University of Science and Technology, Clear Water Bay, Kowloon, 999077 Hong Kong, (SAR), China. ✉e-mail: haipenglu@ust.hk

### (a) Previous strategies of strong photoreductants via two-photon absorption

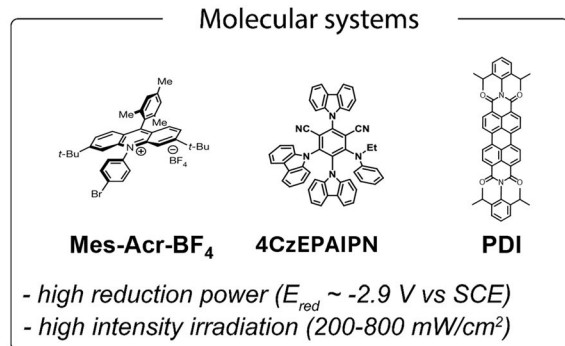

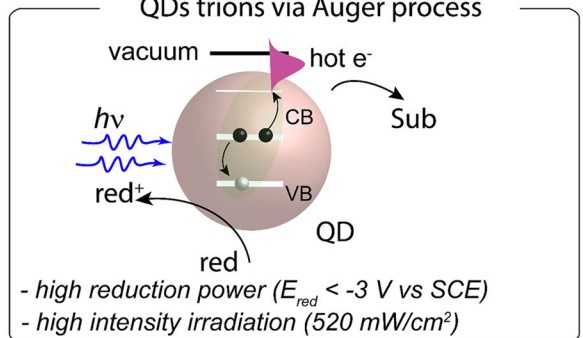

### (b) This work: spin-exchange Auger processes enabling extreme potential photoreduction

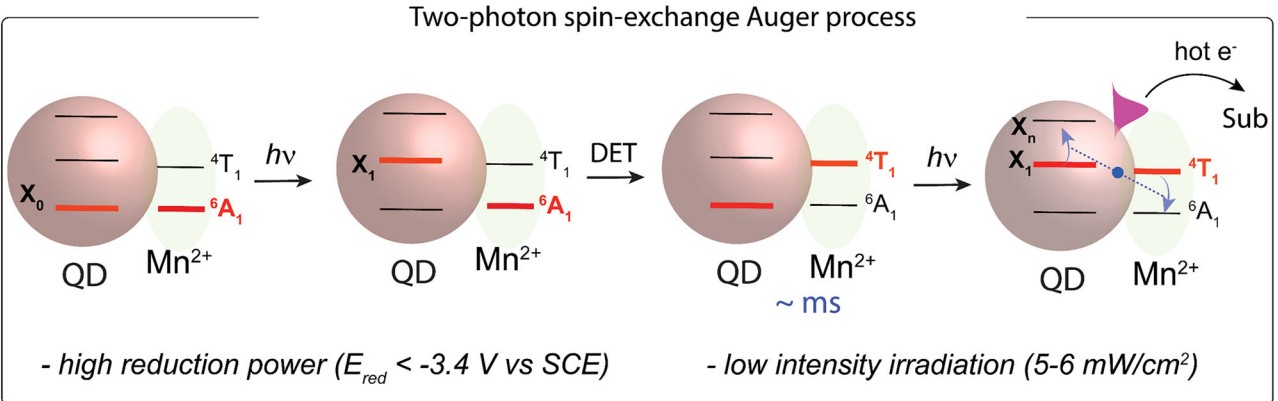

**Fig. 1 | Comparison of strong photoreductants via a two-photon process achieved in previous and present studies.** Top: Scheme of redox potential achieved by single photon process and two photon process; **a** molecular systems (left) and QDs trions via Auger process (right); **b** Extreme potential photoreduction enabled by spin-exchange-enhanced Auger processes.

efficient[3–5]. Furthermore, the redox potential of QDs can be conveniently modulated through synthetic control of nanocrystal size (i.e., quantum confinement[6]) and post-synthetic ligand exchange[7]. Pioneering works from Weiss[8–11], Krauss[12–14], Weix, and Yan[15] have demonstrated that traditional II-VI and $CsPbBr_3$ nanocrystals can drive a wide array of organic transformations with efficiency that rivals those of conventional organic dyes. However, the application of QDs in organic synthesis is still in its infancy compared to small molecule photosensitizers. For instance, most of the demonstrated reactions using QDs can also be achieved with molecular photocatalysts at comparable efficiency. Therefore, the unique capabilities of QDs for photocatalysis warrant further exploration and demonstration.

One crucial advantage of QDs over small molecule photosensitizers, which has been rarely harnessed for photocatalysis, is their ability to generate "hot" electrons and multiple excitons under multiphoton processes with superior photostability compared to molecular dyes[16–18]. These high-energy hot electrons can be produced even with

visible-light irradiation by an "Auger process". The Auger process is an electron-electron Coulomb interaction, where an "energy donor" electron relaxes its energy from the conduction band (CB) to the valence band (VB), while an "acceptor" electron is excited to a higher energy state. This process is greatly enhanced in quantum confined nanocrystals due to the relaxation of the translational momentum conservation rule and the presence of quantized energy levels. If these hot electrons can be effectively harvested, they can serve as potent photoreductants to activate inert substrates with extreme reduction potentials. Traditionally, these molecules are reduced using dissolved alkali metals, posing challenges in synthetic chemistry[19]. Despite recent advancements in generating strong photoreductants using small molecule photosensitizers[20–22] (Fig. 1a, left panel), the need for high catalyst loading and intense near-UV region irradiation remain significant limitations. Therefore, hot electrons enabled by Auger process in QDs present a new opportunity to access these extreme potential substrates with remarkable catalytic efficiency under mild conditions.

Utilizing hot electrons from semiconductors for useful photochemical processes was proposed back in 1978[23,24]; however, it has only been demonstrated lately in chemical reactions. Krauss, Weix and co-workers[12] recently demonstrated that hot electrons from CdS QDs can facilitate a broad range of photoreductive transformations on substrates with reduction potentials reaching up to –3.4 V vs. saturated calomel electrode (SCE). Their mechanistic investigations indicated a two-photon process generating hot electrons by an Auger process through the negative trion state (Fig. 1a, right panel). The upconverted hot electrons possess remarkable reducing power that can activate aryl chlorides and phosphates. This seminal work highlights a catalytic paradigm to access extremely reducing potentials under mild photocatalytic conditions. However, such a two-photon Auger process in QDs typically requires high irradiation fluences (>500 mW/cm²), which is often detrimental to photocatalysis due to the unexpected side reactions and solvent evaporation. On the other hand, it would become much more attractive and convenient if the photocatalytic reaction can proceed under the abundant solar radiation.

Drawing inspiration from recent spectroscopic works[16,25–27], we hypothesize that hot electrons can be more efficiently produced and harnessed in Mn-doped QDs via *sp-d* spin exchange interactions. In this work, we demonstrate that the careful manipulation of the doping chemistry of Mn²⁺ in CdS/ZnS core/shell QDs can indeed modulate the hot-electron yield. The optimized Mn-doped CdS/ZnS QDs (Fig. 1b) produce high-yield and usable hot electrons for a diverse range of extreme-potential photoreductions ($E_{red}$ < –3 V vs SCE), encompassing the Birch reduction, dehalogenation and deoxygenation of arenes, debenzylation, detosylation, and reductive ring openings. Our photocatalyst can perform these reactions under low irradiation power, which is 100 times lower than previous studies in molecular and QD systems. Such a low irradiation requirement is readily achievable under standard solar irradiation conditions (100 mW/cm²). Our work demonstrates the potential of quantum-confined semiconductors in facilitating challenging organic transformations that were unattainable in molecular photocatalysts.

## Results and discussion

### Design and synthesis of Mn-doped core-shell QDs

The concept of harnessing hot electrons for any chemical reaction requires QDs with enhanced relative rates of Auger process compared to other competing processes. Previous femtosecond spectroscopic studies have revealed that in Mn²⁺-doped CdSe QDs, the correlated QDs-Mn²⁺ direct exciton transfer (DET) and Mn²⁺-QDs back exciton transfer (BET) processes through spin-exchange can accelerate the uphill Auger process, surpassing other energy dissipating processes within QDs. Specifically, when incident photons excite the QDs, biexcitons are generated, and one exciton transfers its energy to Mn²⁺ dopants via DET, resulting an Mn²⁺ excited state (⁴T₁). Subsequently, the Mn-to-QDs BET process leads to the Mn²⁺ ground state (⁶A₁) while exciting QDs to a higher excited state. However, in a photocatalytic reaction system, this biexciton mechanism may not be feasible due to the low irradiation fluence of a cw-LED, which cannot generate biexcitons in QDs because of their nanosecond exciton lifetime (see detailed calculation in Supplementary Information Note S2 and Table S1). Here we propose an alternative hot-electron generation mechanism (Fig. 1b) based on a sequential excitation of QDs. In this mechanism, Mn²⁺ dopants act as a temporary energy reservoir, prolonging the exciton lifetime and enabling spin-exchange interactions between excited Mn²⁺ and consecutively generated excitons, resulting in the Auger process under low irradiation intensity. The efficiency of *sp-d* spin-exchange interaction between QD and Mn²⁺ dopants will influence the hot electron yield. The design of QDs aims to map out the precise doping chemistry to maximize the efficiency of spin-exchange interactions.

In this study, we selected CdS as the core QDs due to its larger bandgap compared to CdSe. A typical three-step synthetic method was

employed to precisely control the radial position of Mn²⁺ dopants[28,29] (see Supplementary Information for details). The synthesis and growth of QDs were monitored by UV-vis spectroscopy by examining their exciton peak positions[30]. Initially, we synthesized the CdS core using a previously reported one-pot method[31] and confirmed its phase as zinc blende by powder X-ray diffraction (PXRD) (Fig. S1). The average radius of CdS core was estimated at 1.75 nm based on its first exciton peak at 404 nm (Fig. S2). Subsequently, Mn²⁺ ions were doped into the CdS core using reactive Mn(S₂CNEt₂)₂, which allows dopant incorporation into the lattice at a relatively low temperature. A minor Ostwald ripening effect was observed, leading to a red shift of the exciton peak in Mn²⁺-doped CdS QDs (Fig. S2). Following the doping reaction, the radius of Mn²⁺ doped CdS grew to ~2 nm. To reach the desired bandgap (~2.7 eV), an additional 1.5 monolayers of CdS were grown on Mn²⁺:CdS surface using the successive ionic layer adsorption and reaction (SILAR) method, and the core-doped Mn²⁺:CdS QDs with 2.8 nm radius were obtained. Lastly, a single monolayer of ZnS shell was grown on the Mn²⁺-doped CdS for surface passivation. The CdS/ZnS type I structure facilitate the overlap of hole and electron wavefunctions within the CdS core[32], and minimizes surface traps while enhancing the spin-exchange interactions between the host and Mn²⁺ dopants. The as-synthesized Mn²⁺:CdS/ZnS QDs maintained their zinc blended structure (Fig. S1) and exhibit a narrow size distribution with an average radius of 3.35 nm (Fig. S3). These core-doped QDs are denoted as Mn²⁺_C:CdS/ZnS(as shown in Fig. 2a). For control experiments, we also synthesized Mn²⁺_I:CdS/ZnS, where Mn²⁺ is doped at the core/shell interface (SI).

The incorporation of Mn²⁺ into QDs was first verified through energy-dispersive X-ray spectroscopy (EDS) (Fig. S4) and electron paramagnetic resonance (EPR) (Fig. S5). Both core doped and interface doped CdS/ZnS QDs exhibit a six-line spectrum with a similar hyperfine coupling constant of 193 MHz (corresponding to 69 G), while absent in undoped CdS (Fig. S5). This characteristic hyperfine splitting energy corresponds to substitutional Mn²⁺ ions at the tetrahedral Cd²⁺ sites[28,33]. The concentration of incorporated Mn²⁺ was quantified using inductively coupled plasma atomic emission spectroscopy (ICP-AES, Table S1) to be 0.56% with respect to Cd²⁺. Given the size and lattice parameters of the nanocrystals, we estimated an average of 10 Mn²⁺ dopants per QD (calculation details in Supplementary Information, Note 1). The measured Cd²⁺ to Zn²⁺ ratio was ~1.06, which aligns with the calculation result based on a 2.8 nm of CdS core and 0.55 nm of ZnS shell, indicating a well-defined core/shell system.

The as-synthesized colloidal Mn²⁺_C:CdS/ZnS QDs exhibit two peaks in their PL spectrum (Fig. 2b), a narrow peak at 2.67 eV (464 nm) and a broad peak at 2.1 eV (590 nm). The former peak is attributed to the exciton radiative recombination from CdS, while the latter is ascribed to both Mn²⁺ (⁴T₁-⁶A₁, ~2.1 eV) emission and QDs trap emission (1.9 eV). Time-resolved PL spectroscopy (TRPL) supports these findings. The exciton emission probed at 464 nm shows a characteristic nanosecond lifetime of QDs (Fig. S6a, and Table S2). The PL kinetics probed at 590 nm is composed of microsecond and millisecond decay components (Fig. S6b, and Table S2), indicating two emission mechanisms. The microsecond decay is consistent with trap emissions[34], while the millisecond decay corresponds well to the partially forbidden *d-d* transition of Mn²⁺. This emission from Mn²⁺ indicates a strong exciton-to-Mn²⁺ energy transfer. Compared to the undoped CdS QDs, the PL kinetics shows a faster decay component (Fig. S6, and Table S2), consistent with the DET from host to Mn²⁺ dopants.

To further investigate the energy transfer process in Mn²⁺-doped QDs, we used ultrafast pump-probe transient absorption (TA) spectroscopy (See "methods", SI). QDs were pumped with an *fs*-laser pulse with 3.35 eV photon energy and probed using a white-light continuum. The average exciton number per dot (<*N*>) was ~0.10 (see SI for detailed calculation) to minimize the probability of biexciton

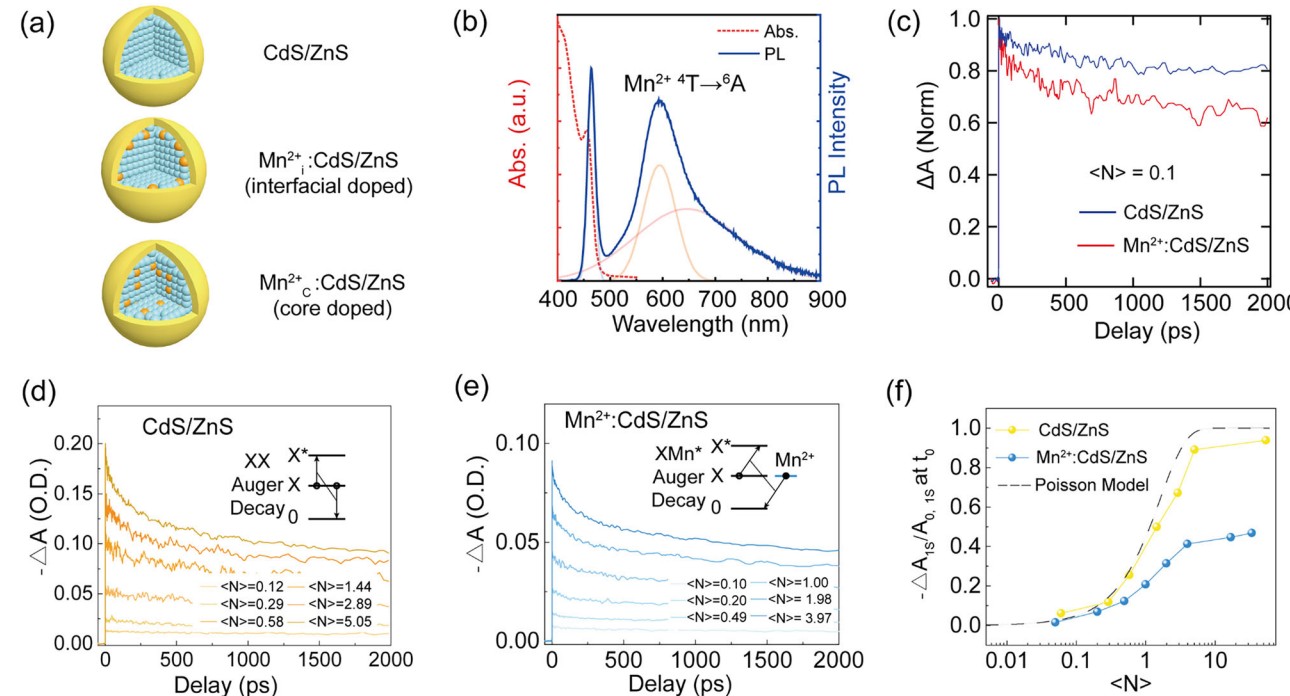

**Fig. 2 | Photophysical properties characterization of as-synthesized QDs.**
**a** Schematics for CdS/ZnS, $Mn^{2+}_i$:CdS/ZnS and $Mn^{2+}_C$:CdS/ZnS QDs; **b** Absorption spectrum (red dash line) and PL spectrum (blue solid line) of colloidal $Mn^{2+}_C$:CdS/ZnS QDs; **c** 1S bleach dynamics of CdS/ZnS, and $Mn^{2+}_C$:CdS/ZnS QDs with a 3.35 eV excitation (<N> -0.1); **d**, **e** Pump-intensity-dependent 1S bleach dynamics of the undoped (**d**) and $Mn^{2+}$-doped CdS/ZnS QDs (**e**). **f** Relative 1S bleach signal at early time for undoped and $Mn^{2+}$-doped CdS/ZnS QDs. The dash line is the Poisson statistics calculated by $n^e_{1S} = 0.5 P(1) + \sum_{i=2}^{\infty} P(i)$.

formation. In both undoped and $Mn^{2+}$-doped CdS/ZnS QDs, the TA signal is dominated by the bandedge (1S) transition bleaching due to filling of the $1S_e$ electron state. The 1S bleach in $Mn^{2+}$-doped CdS/ZnS QDs shows a clearly faster decay kinetics compared to the undoped QDs (Fig. 2c), indicating the ultrafast DET from QDs to $Mn^{2+}$. To extract the time scale of the energy transfer, we employed a subtractive procedure developed by previous studies[35], which accounts for the sample heterogeneity. Fig. S7 shows the difference between the two ΔOD data from the doped and undoped QDs. A single exponential fitting of the difference kinetics yields $\tau_{DET}$ ~ 3 ps for $Mn^{2+}$:CdS/ZnS QDs with 10 $Mn^{2+}$ per dot, which is in close agreement with previous studies[17,35–37]. We expect that the spin exchange process will be further accelerated under high excitation rates.

To probe the dynamics of spin-exchange Auger processes, we further performed fluence-dependent TA experiments. The per-pulse pump fluence is varied to obtain an excitation level of <N> = 0.05 to 54. Figure 2d, e show the dynamics of 1S absorption changes ($\Delta A = A - A_0$, where $A_0$ and A are the absorbance of the ground-state and excited QD, respectively) of undoped and $Mn^{2+}$-doped QDs. As the excitation level increases, the undoped CdS/ZnS QDs display a usual progression of 1S TA dynamics. When <N> is approaching and larger than 1, a clear faster component can be observed in the 1S kinetics owing to the increased biexciton Auger recombination. The time scale of the normal Auger process in undoped CdS/ZnS QDs is estimated as -160 ps (SI, Fig. S8). Interestingly, a qualitatively similar 1S dynamics was observed in $Mn^{2+}_C$:CdS/ZnS QDs. This is different from what Klimov observed in $Mn^{2+}$-doped CdSe/ZnS QDs, where an extremely fast component (200–300 fs) corresponding to the spin-exchange Auger process was observed in 1S dynamics[26,38]. In $Mn^{2+}_C$:CdS/ZnS QDs, the spin-exchange Auger process is largely superimposed with the normal Auger process in the TA kinetics. We attribute this difference to the distinct energy alignments between $Mn^{2+}$ ($^4$T) and the host QDs. In the case of CdSe host, the host's exciton energy is in resonance with $Mn^{2+}$ states, giving rise to an

extremely fast spin-exchange Auger process. However, in CdS QDs, the exciton energy is substantially larger than the $Mn^{2+}$ ($^4$T) state, leading to a slower spin-exchange process. Our observed TA results are also consistent with previous reports of spin-exchange interactions in $Mn^{2+}$-doped CdS QDs[17,35,36].

To confirm this analysis, we plotted the early-time ($t_0$) 1S bleach amplitude ($\Delta\alpha_{1S}$) as a function of the excitation level (Fig. 2f). Early work has shown that the normalized 1S bleach signal ($|\Delta A_{1S}|/A_{0,1S}$) as a function of excitation level should follow the Poisson photon-absorption statistics[33] ($|\Delta A_{1S}|/A_{0,1S} = n^e_{1S} = 0.5 P(1) + \sum_{i=2}^{\infty} P(i)$, where $P(i)$ is the probability of finding $i$ excitons per dot). Figure 2f shows that the normalized 1S bleach signals from undoped QDs follows the Poisson statistics while that of $Mn^{2+}$-doped CdS/ZnS QDs are systematically lower. This suggests that a large fraction of the excited excitons is localized on $Mn^{2+}$ dopants immediately after the photoexcitation. Another important evidence of the spin-exchange Auger process is the derivative feature of TA spectra in $Mn^{2+}$-doped CdS/ZnS QDs. Since the generation of hot electrons would create a charge-separate state, the spatial separation of electrons and holes leads to a strong internal electric field, which will redshift the 1S resonance due to the Stark effect[38]. The derivative feature and the photo-induced absorption become more obvious at higher excitation rates in doped QDs and are absent in undoped QDs at all excitation rates (Fig. S9). This is in agreement with the Poisson analysis and suggests that hot electrons are generated in $Mn^{2+}$-doped CdS/ZnS QDs via an ultrafast spin-exchange Auger process.

## Harnessing hot electrons for extreme-potential photocatalysis: dehalogenation of aryl chlorides as a model reaction

Given that the $Mn^{2+}$-doped CdS/ZnS QDs can produce higher yields of hot electrons than undoped QDs under the same conditions, we envisioned that $Mn^{2+}$ doped QDs can activate inert molecules with high reduction potentials under significantly reduced excitation power compared to the undoped QDs[12]. This approach could help prevent

**Table 1 | Dehalogenation of tert-butyl (4-chlorophenyl) carbamate**

| Entry | Deviation from standard reaction conditions | Yield of 2a[a] | TON |
|---|---|---|---|
| 1[b] | none | 88% | 88000 |
| 2 | no light | 0% | 0 |
| 3 | 1 mW/cm² | 7% | 7000 |
| 4 | 150 mW/cm², 18 h | 93% | 93000 |
| 5 | no TAEA | 8% | 8000 |
| 6 | in air | 13% | 13000 |
| 7 | Toluene instead of DMPU | 7% | 7000 |
| 8 | Mn²⁺$_C$:CdS/ZnS (0.0005 mol%) | 30% | 60000 |
| 9 | CdS/ZnS | 17% | 850 |
| 10 | Mn²⁺$_i$:CdS/ZnS | 16% | 800 |
| 11 | 4-CzIPN (10 mol%) | 19% | 1.9 |
| 12 | [Ir(dFCF₃ppy)₂(dtbbpy)]PF₆ (10 mol%) | 21% | 2.1 |
| 13 | PDI (10 mol%) | 0% | 0 |
| 14 | CdS, 150 mW/cm², 18 h | 21% | 21000 |

[a] NMR yield.
[b] standard condition: 0.25 mmol substrate **1a** was mixed with Mn²⁺$_C$:CdS/ZnS (0.001 mol%) and TAEA (1.5 equiv.) in DMPU (0.25 M) under a N₂ atmosphere. The reaction mixture was then exposed to blue LED irradiation (5 mW/cm²) 55 h.

side reactions and accommodate a broad scope of reactions and substrates with very low catalyst loading.

Hydrodechlorination of *tert*-butyl (4-chlorophenyl) carbamate ($E_{red} = -2.9$ V vs. SCE) was chosen as the model substrate to evaluate the photoreducing capability of our Mn²⁺ doped QDs. This substrate is inert to CdS QDs, as its reduction potential is more negative than the conduction band minimum of CdS QD (−2.24 V vs. SCE[39]). Blue LEDs with a wavelength range of ~455–465 nm (-2.73 eV) was used as the light source. Three light source configurations were examined, with irradiation energy of 1 mW/cm², 5 mW/cm², and 150 mW/cm² measured at the vial position. Tris(2-aminoethyl)amine (TAEA) was used as the sacrificial reductant for the regeneration of QDs. The reaction process was monitored via crude ¹HNMR (Figs. S10, S13) and NMR yields are presented in Table 1. Following condition screening, the optimized reaction condition was determined to including Mn²⁺$_C$:CdS/ZnS QDs (0.001 mol%) as the photocatalyst, *N,N′*-dimethylpropylene urea (DMPU) as the solvent, TAEA as the reductant. We discovered that 5 mW/cm² blue LED irradiation was sufficient to deliver an excellent yield (88%) of the dichlorination product after 55 h of exposure (Table 1, entry 1). A calculated turnover number (TON) as high as 88,000 could be achieved together with a high reaction yield comparable with the literature[12] based on CdS QDs, but with only half the loading of QDs and less than 1% of the irradiation intensity. The reaction did not proceed without light (Table 1, entry 2). Reducing the intensity of blue light to 1 mW/cm² essentially shuts down the dehalogenation (7%, entry 3). Increasing light intensity significantly accelerated the reaction and improved the yield (entry 4)[40]. A very low conversion was observed when TAEA was omitted (8%, Table 1, entry 5). The presence of air negatively impacted the reaction, resulting in a low conversion (Table 1, entry 6), which suggests that the aryl radical intermediate rapidly decomposes in the presence of molecular oxygen. The reaction performed poorly in nonpolar solvents, such as toluene (Table 1, entry 7). Reducing the QD loading to 0.0005 mol% led to a compromised yield (entry 8). When undoped CdS/ZnS or Mn²⁺$_i$:CdS/ZnS QDs were tested, the reaction delivered much lower conversions and TONs (entry 9 and 10). This observation is in line with our TA results, which revealed significantly enhanced spin-exchange interactions in Mn²⁺$_C$:CdS/ZnS QDs, enabling a more efficient Auger process. Several organic dyes and transition metal complexes have been

previously reported as potent photoreductants under two-photon processes[41–43]. These small molecule photoredox catalysts were evaluated in comparison to our Mn²⁺$_C$: CdS/ZnS QDs. As entries 11–13 demonstrate, both organic dyes and Ir complexes resulted in very low reaction yields under low light intensity (5 mW/cm²), even with a high catalyst loading. Minimal TON was achieved, suggesting that the efficiency of the two-photon process was poor under these conditions. Consequently, the spin-exchanged Auger process in Mn-doped QDs exhibits a potential to activate photocatalytically challenging chemical reactions under mild irradiation conditions.

The substrate scope of the dehalogenation of aryl halides was briefly investigated (Fig. 3). The reaction generally exhibited good performance under both 5 mW/cm² and 150 mW/cm² light intensities. To expedite the reaction time, a higher light output (150 mW/cm²) was utilized. Mn²⁺$_C$: CdS/ZnS QDs displayed exceptional tolerance towards a wide range of functional groups. Arenes containing either electron-donating or -withdrawing groups were uniformly well-tolerated (Figs. 3, 1a–k). For substrates containing electron-withdrawing groups, the potentially reducible substituents remained intact. Aryl chlorides possessing reductive potentials as low as −3.4 V vs SCE were successfully dechlorinated using the Mn²⁺$_C$:CdS//ZnS QDs. Hydrodehalogenation of aryl bromides and iodides was also achieved with high efficiency (**1l-1o**).

## Applying the Mn²⁺$_C$:CdS//ZnS QDs to other reductive reactions

The Mn²⁺$_C$:CdS/ZnS QDs were further investigated for their applicability in other challenging photoreductive transformations (Fig. 4). The Birch reduction, which dearomatizes arenes into 1,4-cyclohexadienes, is highly valuable to the pharmaceutical, perfumery industry and academia. Traditional protocols for the Birch reduction necessitate the use of alkali metals in liquid ammonia under cryogenic conditions, posing significant challenges for practical use and scale-up. Without meticulous optimization, we showed that hot electrons generated by Mn²⁺$_C$:CdS/ZnS QDs under 150 mW/cm² irradiation can convert benzene (**7**) to 1,4-cyclohexadienes (**8**) with a 58% reaction yield. Our photocatalyst can successfully perform the Birch reduction at room temperature without the use of any reactive alkali metals or liquid ammonia and only requires solar irradiance.

In addition, the hydrodefunctionalization of variously substituted aryl phosphates was successfully achieved, with yields exceeding 70% (products **4a–c**). Strained C$_{sp3}$-C$_{sp3}$ bonds were also effectively photoreduced using the Mn²⁺$_C$:/ZnS QDs (**6**). The cleavage of sulfonamides represents a valuable yet challenging deprotective method in organic synthesis. Remarkably, both mesyl and tosyl groups were efficiently cleaved under mild visible light irradiation (**10a-10e**). Furthermore, the C$_{sp3}$-O bond cleavage proceeded smoothly for benzyl ether (**11**). Alkyl chloride (**13**) can also be activated to afford the product of reductive dechlorination (**14**) in 32% yield.

## Mechanistic discussion

To study the reaction mechanism, we performed additional control experiments and spectroscopic studies. Previous work[12] showed that a normal Auger process from negative trions of QDs can also produce hot electrons. However, our control experiment shown in Entry 9 from Table 1 indicates that the photodoping process alone is not responsible for the observed high reaction yield in doped QDs. Although QDs can be photodoped by TAEA (Fig. S11), such process is not favourable compared to the ultrafast spin-exchange Auger process. The hole transfer rate from QD* to a hole scavenger is often found to be sluggish and is commonly determined in hundreds of picoseconds or even nanoseconds[44–46]. Therefore, compared to the fast spin-exchange Auger process in Mn²⁺:CdS/ZnS QDs, we expect that the photodoping mechanism is not the main process in generating high-yield hot electrons. The essential role of spin-exchange mechanism is further corroborated with the size-dependent photocatalytic study, where larger

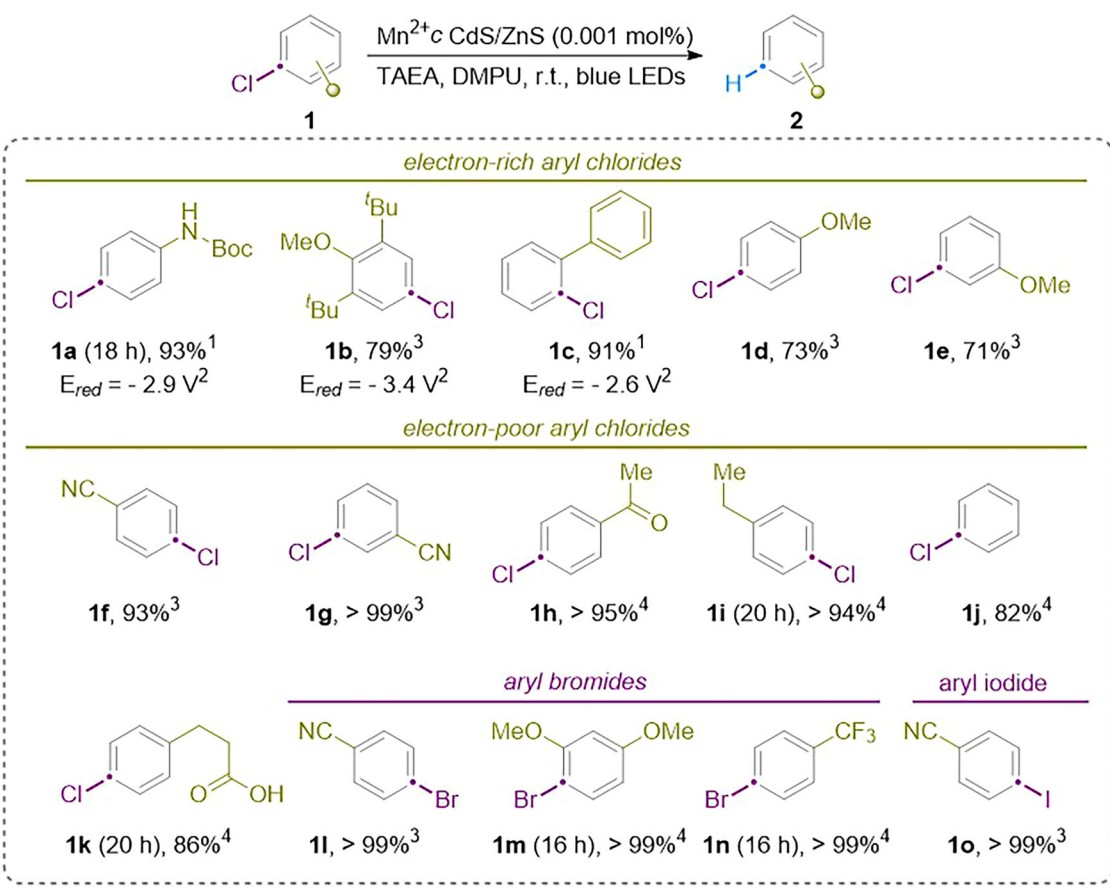

**Fig. 3 | Substrate scope of photodehalogenation of aryl halides catalyzed by Mn²⁺c:CdS/ZnS QDs.** The reactions were performed using the optimized reaction conditions identified in Table 1 with a light intensity of 150 W/cm². [1]Isolated yield. [2]The values of the reduction potential are with respect to SCE. [3]NMR yield. [4]GC yield.

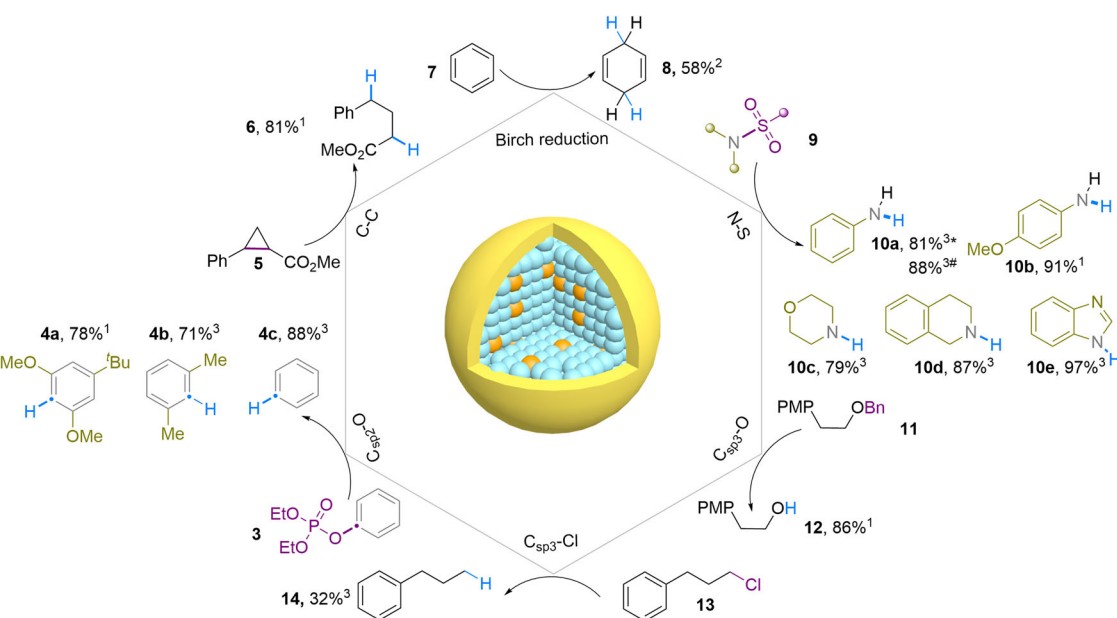

**Fig. 4 | Reductive cleavage of various chemical bonds.** [1]Isolated yield. [2]NMR yield. [3]GC yield *Reduced from N-Phenylmethanesulfonamide # Reduced from N-Phenylbenzenesulfonamide.

CdS core with stronger spin-exchange interaction yields higher reaction yields (Figs. S12, 13).

To confirm the two-photon process, we performed detailed power-dependent photocatalysis study on the model reaction, i.e., the dechlorination of tert-butyl (4-chlorophenyl) carbamate. The production rate shows roughly a linear dependence on the irradiation intensity. Interestingly, Mn²⁺-doped CdS/ZnS QDs exhibits much higher production rate than undoped ones under all irradiation intensities,

further highlighting the importance of spin-exchange Auger process (Fig. S14). The linear dependence of irradiation intensity suggests that our reaction is a sequential two-photo process (Fig. 1b), where intermediate states (i.e., Mn$^{2+}$ excited state $^4$T) quickly build up and reach equilibrium. Thus, the rate order of photons is close to 1. This mechanism is particularly attractive to photocatalysis as it only requires low irradiation intensity to generate hot electrons.

To estimate the likelihood of spin-exchange Auger process, we further calculated the average excitation rate (Note S2). We estimated the average excitation rate for Mn$^{2+}$ ($^4$T) is in the range of 0.11 to 6.8 per QD under the photocatalytic experimental conditions. If we assume that the number of excited Mn$^{2+}$ per QD follows a Poisson distribution (P $(i) = \frac{\lambda^i \times e^{-\lambda}}{i!}$), where $\lambda = <N_{Mn^*}>$ is the average excitation rate of Mn$^{2+}$ state and $i$ is the number of excited Mn$^{2+}$ states per dot, to generate hot electrons, a minimum of $i = 2$ should be obtained. We can see from Table S1, the probability of generating more than two $<N_{Mn}>$ is very high in our photocatalytic system, owing to the extensive lifetime of Mn$^{2+}$ excited state.

Our experiments thus confirmed that the ultrafast spin-exchange Auger process in our Mn$^{2+}$-doped QDs is essential for enabling extreme-potential photoreduction. We proposed the following reaction mechanism. Hot electrons are initially generated from Mn$^{2+}$-doped CdS/ZnS QDs via a *sp-d* spin-exchange Auger process under two-photon irradiation (Fig. 1b), which are subsequently transferred to organic substrates, yielding aryl radicals as key reaction intermediates. The comprehensive catalytic cycles are illustrated in Fig. 5: i) A photoexcited exciton ($i_a$) from the CdS QDs transfers its energy to Mn$^{2+}$ dopants through spin-exchange interactions ($i_b$), resulting in an Mn$^{2+}$ excited state ($^4$T$_1$); ii) a consecutive photon is absorbed to produce the second exciton; iii) Mn$^{2+}$-QDs BET process takes place, exciting the band edge electron to a higher state (with an additional energy of ~2.1 eV) and generating hot electrons, while Mn$^{2+}$ returns to the ground state ($^6$A$_1$); iv) a hot or even a solvated electron generated in step iii is transferred to aryl chloride (single electron transfer, SET), producing positive QDs and aryl radical anions; v) a sacrificial reductant, such as TAEA, donates an electron to the Mn$^{2+}$$_C$:CdS/ZnS QDs, restoring the QDs to their neutral ground state; vi) the aryl radical anion releases a chloride, yielding a highly reactive aryl radical, which is reduced by the TAEA radical cation through hydrogen atom transfer (HAT).

### Performing cross-coupling reactions using aryl halides
The aryl radical generated by the hot electrons can also participate in non-reductive cross-coupling reactions, yielding structurally diverse molecules. Preliminary screening revealed that electron-rich heteroaryls, Ph$_2$S$_2$, B$_2$pin$_2$, and Sn$_2$Me$_6$ were competent radicalophiles capable of intercepting the aryl radical to form C-C, C-S, C-B, and C-Sn bonds

(Figs. 6a, 2p–u). As the Auger process depends on the light intensity of the irradiation, we hypothesized that adjusting the irradiation output could act as a reaction switch to program cascades cross-coupling reactions. Under low light intensity, only bandedge electrons are generated, activating substrates with moderate reduction potential. When the excitation power is increased, the two-photon Auger process becomes more efficient, generating hot electrons to activate more inert substrates. To test this strategy, we examined sequential dehalogenative bifunctionalization using 1-bromo-4-chlorobenzene as the starting material. N-methylpyrrole and B$_2$pin$_2$ were sequentially coupled onto the central benzene ring (Fig. 6b). This highly modular approach potentially offers a general platform for preparing multi-substituted arenes by turning on/off of hot electron generations, which are indispensable building blocks in medicinal chemistry and natural product synthesis.

In this work, we have demonstrated that the spin-exchange Auger process can be significantly enhanced in Mn$^{2+}$-doped QDs, enabling the generation of high-efficiency, usable hot electrons for a wide range of chemical reactions involving substrates with extreme potentials. Our innovative photocatalyst can activate inert organic molecules under remarkably low output (~5 mW/cm$^2$) of visible light, which is 1% of that used in previous demonstrations, while achieving twice the previously reported turnover numbers (TONs). Under these mild experimental conditions, the Mn$^{2+}$-doped QDs exhibit superior catalytic efficiency, and a broader scope of reaction and substrate compatibility compared to molecular photosensitizers. This enhanced performance is attributed to the core-doped Mn$^{2+}$-CdS/ZnS QDs' ability to generate efficient hot electrons due to stronger dopant-host interactions. Our photocatalyst is a significant advance compared to previous ones as it can directly utilize the abundant solar irradiance to activate these inert substrates. Our findings significantly expand the applicability of QDs as photocatalysts for organic synthesis and highlight the immense potential and uniqueness of the Auger process in quantum-confined semiconductors. Furthermore, we have demonstrated the capability of this approach to perform modular cross-coupling cascade reactions that were previously unattainable. This finding holds great promise for the future development of advanced materials and green synthetic chemistry.

## Methods
### Materials
Cadmium nitrate tetrahydrate (Cd(NO$_3$)$_2$·4H$_2$O, Macklin), oleic acid (OA, Aladdin, 90%), oleylamine (OAm, Aladdin, 70%), sodium diethyldithio-carbamate (98%, Bidepharm), manganese acetate anhydrous (Mn(OAc)$_2$, 98%, Aladdin), sodium myristate (>98.0% (GC), Aladdin), 1-octadecene (ODE, 90%, Aladdin), hexane (HPLC, RCI Labscan Ltd.), methanol absolute (VWR Chemicals), acetone (HPLC, RCI Labscan Ltd.), sulfur powder (S, 99.999%, Aladdin), zinc stearate (10-12%, Aladdin).

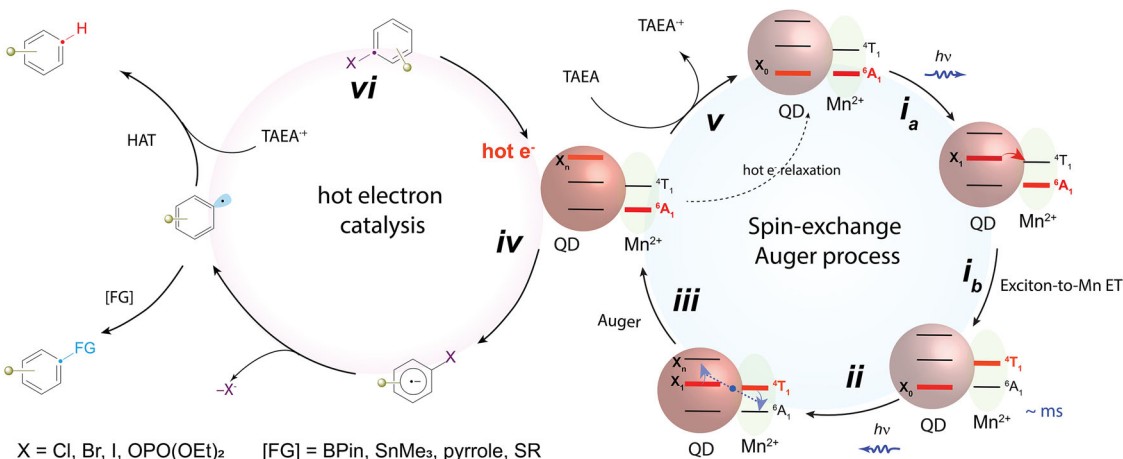

**Fig. 5 | Reaction mechanism.** The proposed catalytic cycles for photoreductive reaction involving the Mn$^{2+}$$_C$:CdS/ZnS QDs.

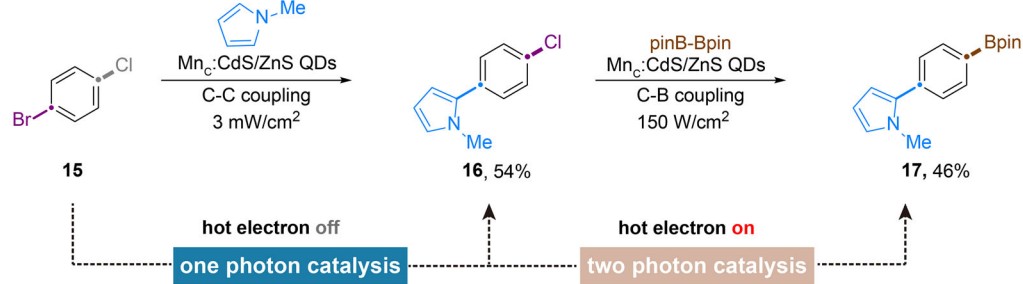

**Fig. 6 | Non-reductive cross-coupling reactions mediated by the aryl radical intermediate. a** reaction scope of one-step cross-coupling reactions; **b** Cascades cross-coupling reactions. Isolated yields were reported.

## Preparation of stock solutions

**Cadmium myristate.** 1 mmol $Cd(NO_3)_2·4H_2O$ (0.3085 g) was dissolved in 20 mL methanol to form 0.05 M cadmium nitrate solution A. 3 mmol sodium myristate was dissolved in 120 mL methanol to form 0.025 M sodium myristate solution B. Solution A was added into solution B under stirring for 15 min. The white precipitants were washed twice with methanol then dried under in vacuum overnight.

**Mn (OAc)$_2$ solution.** OAm was degassed under vacuum at 120 °C for 30 min then cooled down to room temperature. 0.02 mmol $Mn(OAc)_2·4H_2O$ (4.9 mg) was added into 2 mL OAm. The mixture was degassed at room temperature and 120 °C for 10 min for each step. When the desired clear solution was obtained, the solution was cooled to room temperature for further use. It is recommended to use the freshly prepared $Mn^{2+}$ solution.

**Mn(S$_2$CNEt$_2$)$_2$ solution.** 0.44 mmol $NaS_2CNEt_2$ was dissolved in 2 mL degassed OAm under vacuum at room temperature and at 60 °C for 10 min for each step. The obtained oleylamine solution of $NaS_2CNEt_2$ was injected into $Mn(OAc)_2$ solution with stirring under $N_2$ flow. After 10 min, a slightly yellow solution of $Mn(S_2CNEt_2)_2$ was obtained and was used directly for dopant growth. It is recommended to use the freshly prepared $Mn^{2+}$ solution.

**Sulfur solution.** 0.4 mmol sulfur powder (12.8 mg) was mixed with 10 mL ODE. The mixture was degassed at room temperature for 10 min then heated up to 130 °C under $N_2$ flow. After reacting 5 min the solution was cooled to room temperature for use.

**Cadmium myristate solution.** 0.12 mmol Cd(myristate)$_2$ (0.06822 g) and 3 mL ODE were loaded into a three-neck flask, followed by degassing at 120 °C for 10 min. The mixture was kept at 120 °C under $N_2$ flow until all solids were dissolved.

**Zinc-stearate solution.** 0.4 mmol zinc-stearate powder (0.2528 g) and 10 mL ODE were added into a three-neck flask. The mixture was degassed at room temperature for 10 min then 120 °C for 30 min. Next, the mixture was heated to 200 °C under $N_2$ flow to dissolve zinc-stearate powders.

## Synthesis of CdS core QDs

The synthesis of CdS QDs was based on a typical one-pot synthesis of zinc blende CdS method[31]. 0.2 mmol Cd(myristate)$_2$ (113.4 mg) and 0.1 mmol (3.2 mg) S powder were loaded in a 25 mL three necked flask with 10 mL ODE. The mixture was degassed under vacuum at room temperature for 10 min then 120 °C for 20 min. Under $N_2$ flow, the mixture was heated up to 240 °C. The growth was monitored by taking the absorption spectra of aliquots extracted from the reaction solution. When the first exciton peak reached the desired wavelength (403 nm) the reaction was quenched by cooling down to room temperature. The nanocrystals were precipitated by adding acetone. The precipitants were separated by centrifuging at 6580 g for 6 min and then redispersed in hexane.

## Synthesis of undoped CdS/ZnS core/shell QDs

The CdS/ZnS core/shell QDs with ~2.7 eV bandgap was synthesized based on growing CdS "shell" on the CdS core with ~3.0 eV bandgap. 2 mL 50 nmol/mL CdS core dispersed in hexane (concentration is determined from the first exciton peak of Absorption spectrum) was loaded in a 25 mL 3-neck flask with 6 mL ODE and 2 mL OAm. The hexane was removed under vacuum, followed by heating up to 120 °C for 30 min to degas. After degassing, the mixture was heated up to 240 °C under $N_2$ flow. Prepared cadmium myristate solution (0.04 M) and sulfur solution (0.04 M) were alternatively introduced by dropwise addition. Growth time was 10 min after each injection. The size of the nanocrystals was monitored by collecting the absorption spectra. After the first exciton peak reaching 460 nm, the

reaction solution was cooled down to 220 °C. Then zinc-stearate solution (0.04 M) and sulfur solution (0.04 M) were alternatively introduced into the nanocrystal solution by dropwise addition. Growth time was 10 min after each injection. After the desired shell thickness was achieved, the reaction solution was heated to 280 °C and kept for 5 min. Then the reaction was stopped by cooling down to room temperature. The nanocrystals were precipitated by adding acetone. The precipitants were separated by centrifuging at 8330 g for 6 min and then redispersed in hexane.

### Synthesis of core-doped Mn$^{2+}_C$:CdS/ZnS QDs

60 nmol Mn$^{2+}$ doped CdS core was loaded into 50 mL 3-neck flask with 6 mL ODE and 2 mL OAm. After removing the hexane and degassing under vacuum at 120 °C, the mixture was heated to 240 °C under N$_2$ flow. Prepared cadmium myristate solution (0.04 M) and sulfur solution (0.04 M) was alternatively introduced by dropwise addition. Growth time was 10 min after each injection. The size of the nanocrystals was monitored by collecting the absorption spectra. After the first exciton peak reaching 460 nm, the reaction solution was cooled down to 220 °C. Growth of ZnS shell: Then zinc-stearate solution (0.04 M) and sulfur solution (0.04 M) were alternatively introduced into the nanocrystal solution by dropwise addition. Growth time was 10 min after each injection. After the desired shell thickness was achieved, the reaction solution was heated to 280 °C and kept for 5 min. Then the reaction was stopped by cooling down to room temperature. The nanocrystals were purified by adding acetone and then centrifuged at 10280 g for 7 min. The resulting nanocrystals were redispersed in hexane.

### Synthesis of interface-doped Mn$^{2+}_I$:CdS/ZnS QDs

60 nmol Mn$^{2+}$ doped CdS core (with a diameter of 5.7 nm) was loaded into 50 mL 3-neck flask with 6 mL ODE and 2 mL OAm, followed by removing the hexane and degassing under vacuum at 120 °C. After degassing, the mixture was heated up to 220 °C under N$_2$ flow. Then zinc-stearate solution (0.04 M) and sulfur solution (0.04 M) were alternatingly introduced into the nanocrystal solution by dropwise addition. Growth time was 10 min after each injection. After the desired shell thickness was achieved, the reaction solution was heated to 280 °C and kept for 5 min. Then the reaction was stopped by cooling down to room temperature. The nanocrystals were purified by adding acetone and then centrifuged at 10280 g for 7 min. The resulting nanocrystals were redispersed in hexane.

### Characterization method

**Powder X-ray diffraction (XRD)** patterns are recorded on an Empyrean PANanalytical diffractometer with an ADDS wide-angle X-ray powder diffractometer (Cu Kα radiation, λ = 1.54184 Å). **Nuclear Magnetic Resonance (NMR)** are collected on a Bruker AVII 400 MHz NMR Spectrometer. This spectrometer is equipped with a PA BBO 400SB BBFO-H-D05 Z-gradient BB observe probe head, BB = $^{19}$F-$^{31}$P-$^{15}$N; with Z-gradient with active shielding for 5 mm and ATM accessory. **UV-vis absorption spectroscopic** measurements are performed using a UH5700 spectrophotometer. The scans are performed for a wavelength range of 380 nm – 550 nm, with 1 nm step size and repetition of one cycle. The nanoplatelets sample is dispersed in toluene and loaded in a 10 mm × 10 mm quartz cuvette in colloidal form. **Photoluminescence (PL) spectra** are recorded using a fluorescence spectrometer (FS5, Edinburgh) equipped with a xenon lamp, the excitation wavelength is 360 nm. **The Transmission electronic microscopy characterization (TEM)** are collected by JEM 2010F (JEOL) and JEM-ARM200F (JEOL). **Inductively Coupled Plasma –Optical Emission Spectrometer (ICP−OES)** results are collected by AVIO-200 (Perkin Elmer). **Gas chromatography-Flame ionization detection (GC-FID)** data are collected on 5975 C inert MSD with Triple-Axis Detector (Agilent Technologies). X-band (9.35 GHz) continuous-wave **electron paramagnetic resonance (EPR)** spectra at 296 K were recorded on a Bruker EleXsys E500 spectrometer equipped with a super-high Q resonator (ER4122SHQE) in the Department of Chemistry at the Southern University of Science and Technology (SUSTech). Experimental parameters are as following: microwave frequency = 9.35 GHz; microwave power = 0.02 mW; conversion time = 40 ms; modulation amplitude = 0.5 mT; modulation frequency = 100 kHz.

## Data availability

Data available with the paper or supplementary information. Data supporting the findings of this manuscript are also available from the corresponding author upon request.

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

## Acknowledgements

We gratefully thank Kaifeng Wu and Jianwei Sun for their insightful discussion. This work was supported by the Research Grants Council of Hong Kong via the General Research Fund (16300123), Collaborative Research Fund (C1055-23G), the National Natural Science Foundation of China (Grant No. 22205186 and No. NSFC-RGC 22461160285 (Y. Y.)) and NSFC-RGC Joint Research Scheme (N_HKUST616/24). We also acknowledge the start-up funding support from the Hong Kong University of Science and Technology (HKUST) School of Science (SSCI) and the Department of Chemistry (R9270).

## Author contributions

Q.C. and H.L. conceived the ideas and designed the project. Q.C. synthesized and characterized the samples and performed photocatalysis. J.F. and S.Z. help with the photocatalysis. K.F., J.Z., J. X., X.L., and K.W. carried spectroscopic measurements. A. S., Y. Y., K. S.W., and Y.H. participated in discussions. B.M. carried EPR measurements under the supervision of L.T. Q.C., Y.H. and H.L. wrote the manuscript with contributions from all authors.

## Competing interests

The authors declare no competing interests.
