## [Transparent Peer Review file · Nature Communications]

Extreme Potential Photocatalysis Enabled by Spin-Exchange Auger Processes in Magnetic-Doped Quantum Dots

Corresponding Author: Professor Haipeng Lu

Version 0:

Reviewer comments:

Reviewer #2

(Remarks to the Author)

In this revised manuscript the authors report on photocatalysis using Mn²⁺ doped colloidal quantum dots, with the main advance being a highly reducing electron can be photochemically produced from a spin-exchange Auger process. (Note that in terms of impact while AM1.5 solar flux is important for solar fuels/energy applications, when making chemical bonds of the nature described here, it is unclear why the standard commercial LEDs are not sufficient). Regardless, the demonstration of cross coupling reactions and the Birch reduction (Tables 3 and 4) are very exciting and important observations worthy of publication. On the other hand, while the authors have gone a long way to address the concerns of the reviewers in the original version, there still seems to be some ambiguity and inconsistencies regarding the spin-exchange mechanism. Specifically,

1) In prior demonstrations of the spin-exchange effect by Klimov and co-workers (Nature Photonics volume 16, pages 433–440 (2022)) the TA shows a few hundred femtosecond decay due to the spin-exchange effect. The authors do not see this fast component but try to explain it away as being too fast for their instrument response. This is not an acceptable explanation for why the signal was not observed. The spin exchange should be observable directly in TA even with a 100 fs instrument response. Absent this fast component it is suggestive of another two-photon process for creating the highly reducing electron. Is there a better explanation for the TA data?

2) In the same Nat Photonics paper it was shown that the PL lifetime $\tau_{PL} \approx (1 - f) - 1\tau_X$, where τ_X is the exciton lifetime and f = fraction of excitons on Mn. This is inconsistent with the PL lifetime data presented here in two ways:

- a) the doped QD lifetime should be longer (not shorter) and
- b) the doped QD lifetime should not have millisecond components.

Why is the PL lifetime data not consistent with earlier reports (in CdSe) of the spin-exchange effect?

3) It is not clear that photodoping can be eliminated completely as a possible contributing factor. Photodoping creates a steady state population of reduced quantum dots (J. Am. Chem. Soc. 2013, 135, 50, 18782–18785) even under very mild excitation conditions, and even in absence of reductants (J. Phys. Chem. C 2021, 125, 41, 22650–22659). Can photodoping be eliminated conclusively?

4) The spin-exchange effect relies on a relatively good alignment of the Mn 4T₁-6A₁ transition with the bandgap of the QD. Is it possible to change the size of the CdS core to explore this further, which could help to rule in or out the spin-exchange mechanism?

5) The CdS Core QDs look really triangular, and thus not spherical, which would impact the accuracy of the calculations of “core” and “shell” size. For example, it is unlikely that there is 0.55 ML of a ZnS shell. The error in thickness is much larger than this, so is it possible that this uncertainty could impact the photochemistry and doping, etc. for such a thin shell.

Reviewer #3

(Remarks to the Author)

The main issue I have with this manuscript is the following statement:

“Drawing inspiration from recent spectroscopic works,16,25-26,27 we hypothesize that hot electrons can be more efficiently produced and harnessed in Mn-doped QDs via sp-d spin exchange interactions. In this work, we demonstrate that the careful manipulation of the doping chemistry of Mn²⁺ in CdS/ZnS core/shell QDs can indeed modulate the hot-electron yield.”

The authors' central hypothesis that serves as the foundation of this work has already been validated by Son and coworkers in their photoelectron emission studies correlated with QD structure. These studies, along with applications in hot electron-enhanced reactions such as HER, CO₂ reduction, and the redox-neutral conversion of formic acid to CO, cited in the manuscript, provide evidence for the phenomenon and catalytic value. Additionally, mechanistic insights from the groups of Son and coworkers and Klimov and coworkers further support this understanding.

Given the substantial body of existing literature, the novelty of the methodology presented is not as significant as the authors claim. While the application of hot electrons to the reaction is solid, I am unable to recommend this manuscript for publication in Nature Communications for the reasons stated above.

Version 1:

Reviewer comments:

Reviewer #2

(Remarks to the Author)

In the revised version of the manuscript, the authors have tried to address the concern regarding the dynamics of the spin-Auger process that presented itself in the first version. They propose that CdS is fundamentally different from CdSe in that regard. This may or may not be true, but certainly beyond the scope of the manuscript to figure it all out. (Sounds like a nice follow up study).

Publication is enthusiastically recommended.

We thank the reviewers for their meticulous examination of our manuscript and for providing valuable suggestions, inquiries, and comments. We have taken into account all reviewer feedback and provided a detailed point-by-point response below. The comments of the reviewers are presented in italics, followed by our response in plain text. Any new content incorporated into the manuscript is distinguished by **blue text**. We are grateful for the reviewers' insightful feedback, which has significantly improved the quality of our work.

Reviewer #2:

Comments:

In this revised manuscript the authors report on photocatalysis using Mn²⁺ doped colloidal quantum dots, with the main advance being a highly reducing electron can be photochemically produced from a spin-exchange Auger process. (Note that in terms of impact while AM1.5 solar flux is important for solar fuels/energy applications, when making chemical bonds of the nature described here, it is unclear why the standard commercial LEDs are not sufficient). Regardless, the demonstration of cross coupling reactions and the Birch reduction (Tables 3 and 4) are very exciting and important observations worthy of publication. On the other hand, while the authors have gone a long way to address the concerns of the reviewers in the original version, there still seems to be some ambiguity and inconsistencies regarding the spin-exchange mechanism. Specifically,

Response: We thank the reviewer for acknowledging the significance of this work. We are grateful for his/her continuous support and insightful feedback, which has significantly improved the quality of our work. Below we provide a *point-by-point* response to address his/her comments.

1) In prior demonstrations of the spin-exchange effect by Klimov and co-workers (Nature Photonics volume 16, pages 433–440 (2022)) the TA shows a few hundred femtosecond decay due to the spin-exchange effect. The authors do not see this fast component but try to explain it away as being too fast for their instrument response. This is not an acceptable explanation for why the signal was not observed. The spin exchange should be observable directly in TA even with a 100 fs instrument response. Absent this fast component it is suggestive of another two-photon process for creating the highly reducing electron. Is there a better explanation for the TA data?

Response: We thank the reviewer for the critical comment. We would like to highlight that the spin-exchange Auger process in our Mn²⁺-doped CdS/ZnS QDs was unambiguously supported by the following spectroscopic and photocatalysis results:

- a. Mn²⁺ emission was observed by exciting the CdS/ZnS QDs host, confirming the spin-exchange-mediated energy transfer process from QDs to Mn²⁺ dopants.
- b. Spectroscopic features of hot electrons were observed from TA spectra at high excitation rates in Mn²⁺-doped CdS/ZnS QDs, confirming the generation of hot electrons.

- c. The early-time (t_0) 1S bleach amplitudes (ΔA_{1S}) of Mn^{2+} -doped CdS/ZnS QDs were lower than the predicted Poisson distribution at higher excitation level, suggesting an ultrafast process occur at early time.
- d. Hot-electron photocatalysis was enabled only with Mn^{2+} -doped CdS/ZnS QDs and not with the undoped QDs.

However, we do not observe the as fast spin-exchange Auger process as that in Klimov's work.

We agree with the reviewer that it should be observable in TA even if the process is within the instrument response. Therefore, we have revised our discussion accordingly. We believe that the spin-exchange Auger process in our Mn^{2+} -doped CdS/ZnS QDs is actually slower than in Mn^{2+} -doped CdSe/ZnS QDs (as shown in Klimov's work). Additionally, the spin-exchange Auger process is largely superimposed with the normal Auger process. As such, we could not see a distinct spin-exchange Auger process from TA kinetics in Mn^{2+} -doped CdS/ZnS QDs.

We attribute the faster spin-exchange Auger process in Mn^{2+} -doped CdSe/ZnS QDs to the energy resonance between the excitons of CdSe QDs and the Mn^{2+} excited state (Figure R1). In Mn^{2+} -doped CdS/ZnS QDs, the exciton energy is not in resonance with the Mn^{2+} excited state; therefore, the spin-exchange interaction is significantly slower.

Figure R1. Schematic depiction of the spin-exchange interactions in Mn^{2+} -doped CdSe (a) and Mn^{2+} -doped CdS (b) QDs.

Our TA results are also consistent with previous TA studies on Mn^{2+} -doped CdS/ZnS QDs (*J. Phys. Chem. C* 2010, 114, 4418; *J. Phys. Chem. C* 2012, 116, 9300; *J. Phys. Chem. C* 2011, 115, 11407), where compelling evidence were shown for the spin-exchange Auger process. However, none of these studies showed any features as fast as what Klimov observed in Mn^{2+} -doped CdSe/ZnS QDs. Therefore, the spin-exchange interaction in Mn^{2+} :CdS is fundamentally different from that in Mn^{2+} :CdSe QDs due to the distinct energy alignments.

Revision:

We have revised the TA discussion accordingly.

“In Mn^{2+} :CdS/ZnS QDs, the spin-exchange Auger process is largely superimposed with the normal Auger process in the TA kinetics. We attribute this difference to the distinct energy alignments between Mn^{2+} (4T) and the host QDs. In the case of CdSe host, the host's exciton

energy is in resonance with Mn^{2+} states, giving rise to an extremely fast spin-exchange Auger process. However, in CdS QDs, the exciton energy is substantially larger than the Mn^{2+} (^4T) state, leading to a slower spin-exchange process. Our observed TA results are also consistent with previous reports of spin-exchange interactions in Mn^{2+} -doped CdS QDs (*J. Phys. Chem. C* 2010, 114, 4418; *J. Phys. Chem. C* 2012, 116, 9300; *J. Phys. Chem. C* 2011, 115, 11407). ”

2) In the same *Nat Photonics* paper it was shown that the PL lifetime $\tau_{\text{PL}} \approx (1 - f) - 1\tau_X$, where τ_X is the exciton lifetime and f = fraction of excitons on Mn. This is inconsistent with the PL lifetime data presented here in two ways:

a) the doped QD lifetime should be longer (not shorter) and

b) the doped QD lifetime should not have millisecond components.

Why is the PL lifetime data not consistent with earlier reports (in CdSe) of the spin-exchange effect?

Response: We thank the reviewer for the critical comment. As illustrated in Response #1 and Figure R1, the photophysics of Mn^{2+} :CdS is fundamentally different from that of Mn^{2+} :CdSe QDs. In Mn^{2+} :CdSe QDs, the host's exciton energy is in resonance with Mn^{2+} states, that is, the energy of host exciton can be transferred forward and back to Mn^{2+} states. Since the Mn^{2+} excited state (^4T) has a longer radiative lifetime, the Mn^{2+} :CdSe QDs would display a longer lifetime than undoped one, akin to the thermally activated delayed fluorescence (TADF).

However, in the case of Mn^{2+} :CdS, the host's exciton energy is **not** in resonance with that of Mn state. Therefore, there is only a unidirectional energy transfer from QDs to Mn dopants and the doped QDs would display a shorter radiative lifetime than the undoped ones. Please note that our TRPL result of Mn^{2+} :CdS is also consistent with literature reports *J. Am. Chem. Soc.* 2025, 147, 9, 7965; *Chem. Eur. J.* 2009, 15 (13), 3186; *ACS Nano* 2017, 11 (12), 12591.

3) It is not clear that photodoping can be eliminated completely as a possible contributing factor. Photodoping creates a steady state population of reduced quantum dots (*J. Am. Chem. Soc.* 2013, 135, 50, 18782–18785) even under very mild excitation conditions, and even in absence of reductants (*J. Phys. Chem. C* 2021, 125, 41, 22650–22659). Can photodoping be eliminated conclusively?

Response: We thank the reviewer for the comment. We agree with the reviewer that the photodoping mechanism may not be fully excluded completely. However, photodoping alone, is not responsible for the success of our reactions. This is proved by the control experiments (entry 9 in Table 1) where undoped CdS/ZnS cannot generate efficient hot electrons under our experimental conditions to activate the substrates.

4) The spin-exchange effect relies on a relatively good alignment of the Mn 4T1-6A1 transition with the bandgap of the QD. Is it possible to change the size of the CdS core to explore this further, which could help to rule in or out the spin-exchange mechanism?

Response: Yes, it is an excellent point! It is critical that the bandgap is in good alignment of the Mn transition to have a strong spin-exchange effect. We have varied the size of CdS core and performed additional photocatalytic reactions. Our results show that the size of CdS core indeed plays a critical role in the performance of photocatalysis. Our size-dependent photocatalytic study shows that larger CdS core gives higher reaction yield. This is because larger CdS core leads to smaller bandgaps, which display a better energy alignment with the Mn transition. Our results also show that when the Mn²⁺ emission is weak, in the case of weak spin-exchange interaction, the reaction yield appears to be low. We can only obtain high reaction yields when the Mn²⁺ emission is strong under strong spin-exchange interaction. Therefore, our experiments further confirm the mechanism of spin-exchange Auger mechanism.

Figure R3. (a) Size dependent absorption spectra; (b) PL spectrum of Mn²⁺:CdS/ZnS QDs with bandgap of 2.77 eV; (c) PL spectrum of Mn²⁺:CdS/ZnS QDs with bandgap of 2.70 eV; (d) PL spectrum of Mn²⁺:CdS/ZnS QDs with bandgap of 2.62 eV.

Figure R4. Reaction yields afforded by Mn²⁺:CdS/ZnS QDs photocatalysts with different size.

Revision:

“Therefore, compared to the fast spin-exchange Auger process in Mn²⁺:CdS/ZnS QDs, we expect that the photodoping mechanism is not the main process in generating high-yield hot electrons. The essential role of spin-exchange mechanism is further corroborated with the size-dependent photocatalytic study, where larger CdS core with stronger spin-exchange interaction yields higher reaction yields (Figure S12-13).”

5) The CdS Core QDs look really triangular, and thus not spherical, which would impact the accuracy of the calculations of “core” and “shell” size. For example, it is unlikely that there

is 0.55 ML of a ZnS shell. The error in thickness is much larger than this, so is it possible that this uncertainty could impact the photochemistry and doping, etc. for such a thin shell.

Response: We thank the reviewer for the critical comment. Please note that we do not have a 0.55 ML of ZnS shell. In the manuscript, we stated that the thickness of ZnS shell is 0.55 nm, which is roughly one monolayer.

Reviewer #3 (Remarks to the Author):

The main issue I have with this manuscript is the following statement:

“Drawing inspiration from recent spectroscopic works,^{16,25-26,27} we hypothesize that hot electrons can be more efficiently produced and harnessed in Mn-doped QDs via sp-d spin exchange interactions. In this work, we demonstrate that the careful manipulation of the doping chemistry of Mn²⁺ in CdS/ZnS core/shell QDs can indeed modulate the hot-electron yield.”

The authors' central hypothesis that serves as the foundation of this work has already been validated by Son and coworkers in their photoelectron emission studies correlated with QD structure. These studies, along with applications in hot electron-enhanced reactions such as HER, CO₂ reduction, and the redox-neutral conversion of formic acid to CO, cited in the manuscript, provide evidence for the phenomenon and catalytic value. Additionally, mechanistic insights from the groups of Son and coworkers and Klimov and coworkers further support this understanding.

Given the substantial body of existing literature, the novelty of the methodology presented is not as significant as the authors claim. While the application of hot electrons to the reaction is solid, I am unable to recommend this manuscript for publication in Nature Communications for the reasons stated above.

Response: We thank the reviewer for the comment. We understand that there are previous works reporting on the hot electron generation through the spin-exchange process in Mn²⁺-doped QDs, and we have cited those works. We also acknowledge that previous works have “provide evidence for the phenomenon and catalytic value”. However, their potential in organic synthesis is far from explored. The literature mentioned by reviewer mostly dealt with simple electron transfer process and have not been exploited in sophisticated chemical bond formation and cleavage. A concept will always remain as a concept until it is fully demonstrated.

Here, we demonstrate that this fascinating photophysical process can indeed be used for many challenging organic transformations including the Birch reduction and reductive cleavage of C-Cl, C-Br, C-I, C-O, C-C, and N-S bonds. We believe that our work has showcase the incredible value of hot electron photocatalysis using quantum-confined semiconductors. We envision that our work will inspire researchers not only in materials chemistry, but also in organic chemistry, to explore new reaction paradigms using colloidal quantum dots.